# Intracellular Toxic Advanced Glycation End-Products May Induce Cell Death and Suppress Cardiac Fibroblasts

**DOI:** 10.3390/metabo12070615

**Published:** 2022-07-01

**Authors:** Takanobu Takata, Akiko Sakasai-Sakai, Masayoshi Takeuchi

**Affiliations:** 1Department of Advanced Medicine, Medical Research Institute, Kanazawa Medical University, Uchinada 920-0293, Ishikawa, Japan; asakasai@kanazawa-med.ac.jp (A.S.-S.); takeuchi@kanazawa-med.ac.jp (M.T.); 2Department of Life Science, Medical Research Institute, Kanazawa Medical University, Uchinada 920-0293, Ishikawa, Japan

**Keywords:** cardiovascular disease, advanced glycation end-products, glyceraldehyde, toxic AGEs, human cardiac fibroblasts

## Abstract

Cardiovascular disease (CVD) is a lifestyle-related disease (LSRD) induced by the dysfunction and cell death of cardiomyocytes. Cardiac fibroblasts are activated and differentiate in response to specific signals, such as transforming growth factor-β released from injured cardiomyocytes, and are crucial for the protection of cardiomyocytes, cardiac tissue repair, and remodeling. In contrast, cardiac fibroblasts have been shown to induce injury or death of cardiomyocytes and are implicated in the pathogenesis of diseases such as cardiac hypertrophy. We designated glyceraldehyde-derived advanced glycation end-products (AGEs) as toxic AGEs (TAGE) due to their cytotoxicity and association with LSRD. Intracellular TAGE in cardiomyocytes decreased their beating rate and induced cell death in the absence of myocardial ischemia. The TAGE levels in blood were elevated in patients with CVD and were associated with myocardial ischemia along with increased risk of atherosclerosis in vascular endothelial cells in vitro. The relationships between the dysfunction or cell death of cardiac fibroblasts and intracellular and extracellular TAGE, which are secreted from certain organs, remain unclear. We examined the cytotoxicity of intracellular TAGE by a slot blot analysis, and TAGE-modified bovine serum albumin (TAGE-BSA), a model of extracellular TAGE, in normal human cardiac fibroblasts (HCF). Intracellular TAGE induced cell death in normal HCF, whereas TAGE-BSA did not, even at aberrantly high non-physiological levels. Therefore, only intracellular TAGE induced cell death in HCF under physiological conditions, possibly inhibiting the role of HCF.

## 1. Introduction

Cardiovascular disease (CVD) is a lifestyle-related disease (LSRD) caused by the dysfunction and cell death of cardiomyocytes via ischemic or non-ischemic events [1,2,3]. Cardiac fibroblasts are crucial for the repair and remodeling of the heart. When injured, cardiomyocytes release cell-specific signaling molecules, such as cytoplasmic contents, cytokines, including interleukin (IL)-1β, IL-6, and tumor necrosis factor-α, and growth factors, such as transforming growth factor (TGF)-β, which activate cardiac fibroblasts and induce their differentiation into cardiac myofibroblasts. Activated cardiac fibroblasts and cardiac myofibroblasts secrete connective tissue growth factor to promote antioxidant effects in cardiomyocytes [3], and proteins, such as collagen, matrix metalloproteinases (MMPs), and tissue inhibitors of MMPs to protect the cardiac tissue integrity [3,4,5]. In contrast, angiotensin II and TGF-β secreted from activated cardiac fibroblasts promote the production of reactive oxygen species by nicotinamide adenine dinucleotide phosphate (NADPH) oxidase and induce apoptosis in cardiomyocytes [3]. Moreover, excessive secretion of collagen, connective tissue growth factor, angiotensin II, and TGF-β induce fibrosis in cardiac tissue [4,5,6]. Cardiac fibroblasts exert similar effects in other diseases, such as cardiac hypertrophy [4,5,6]. Hence, cardiac fibroblasts have a dual role. We designated glyceraldehyde (GA, a product of glucose and fructose metabolism)-derived advanced glycation end-products (AGEs) as toxic AGEs (TAGE) due to their enhanced cytotoxic effect compared to other AGEs as well as their association with LSRD [2]. Instead of glycosylation, TAGE is generated by a non-enzymatic reaction between GA and proteins [7]. We recently revealed that intracellular TAGE decreased the rates of beating and induced cell death in cardiomyocytes in the absence of myocardial ischemia [8]. However, it currently remains unclear whether extracellular TAGE (e.g., TAGE in the blood) directly induces cytotoxicity against cardiomyocytes. Although the organs that generate and release TAGE into the blood have been identified, we revealed that the blood levels of TAGE were elevated in patients with myocardial ischemia, and correlated with risk factors for CVD, as well as promoted the generation of intracellular reactive oxygen species and activation of nuclear factor-κβ in vascular endothelial cells in vitro [2]. This may result in the induction of atherosclerosis. Although “dietary AGEs” are consumed in the daily diet and are present in the blood, “dietary TAGE” do not possibly exist, as they were not detected in over 1600 commercial foods and beverages [2]. Therefore, TAGE are generated in the organs and released into the blood. The relationships between the dysfunction or cell death of cardiac fibroblasts and Intra-/extracellular TAGE remain unclear. GA is generated from glucose and fructose in the heart via three pathways that require aldose reductase, sorbitol dehydrogenase, ketohexokinase, and aldolase B [2,9,10]. The cardiac fibroblasts account for approximately 60–70% of cardiac cells [3]. Based on these findings, we hypothesized that intracellular TAGE is generated in cardiac fibroblasts, which induces their dysfunction or death, while extracellular TAGE may exhibit cytotoxicity via the receptor for AGEs (RAGE) [11]. In brief, we speculated that TAGE suppresses the normal heart functioning. Therefore, we investigated whether intracellular TAGE and TAGE-modified bovine serum albumin (BSA) (TAGE-BSA), a model of extracellular TAGE, induces cell death in normal human cardiac fibroblasts (HCF).

## 2. Results

### 2.1. Viability Based on NADPH Activity and Accumulated Levels of Intracellular TAGE in GA- and Aminoguanidine (AG, an Inhibitor of AGE Production, the Amino Group of Which Reacts with the Carbonyl Group of Carbohydrates)-Treated HCF

The viability of GA-treated HCF decreased whereas that of intracellular TAGE increased in a GA dose-dependent manner (Figure 1a,b). The viability of HCF treated with 16 mM AG decreased to 76%. The viabilities of HCF treated with 16 mM AG followed by 1 and 2 mM GA decreased to 75 and 77%, respectively (Figure 1c). Since no significant differences were observed in the viabilities of cells treated with AG followed by 0, 1, and 2 mM GA treatment, we suggest that AG completely suppressed cell death induced by GA. In the absence of the AG pretreatment, the accumulation of intracellular TAGE increased in a GA dose-dependent manner from 0 to 2 mM GA; however, these increments were completely inhibited in HCF pretreated with AG (Figure 1d). 

### 2.2. Cell Viability Based on the NADPH Activity of HCF Treated with Non-Glycated-BSA (NG-BSA) and TAGE-BSA (the Model of Extracellular TAGE)

We examined the cytotoxicity of TAGE-modified proteins in HCF, and the concentration of NG-BSA used was the same as that of TAGE-BSA which was used as the model of extracellular TAGE. To confirm the possibility that amino acid sequences and structure of BSA affect HCF, we treated NG-BSA in the same manner as TAGE-BSA. The cytotoxicity of TAGE-BSA was compared to that of the control and NG-BSA groups. In comparison with the control, which contained phosphate-buffered saline (PBS) without calcium and magnesium (PBS)(−), 100 and 200 μg/mL of NG-BSA and TAGE-BSA did not induce cell death or affect cell growth (Figure 2). In contrast, the viability of HCF treated with TAGE-BSA did not significantly differ from that of those treated with NG-BSA and (PBS)(−). These findings suggest that TAGE does not exhibit cytotoxicity or affect cell growth.

## 3. Discussion

In the present study, intracellular TAGE was generated by GA-treated HCF (Figure 1b,d). In this method, treatment with GA effectively reflected physiological conditions. Since the treatment with 2 mM GA corresponded to a glucose concentration of 20 mM in the pancreatic islets [8,12,13], the conditions for the cell culture in an 8 mM glucose-supplemented medium with 2 mM GA may be replaced by 28 mM glucose. 

Collectively, the results showed that intracellular TAGE strongly induces cell death (Figure 1a,b). To confirm this, HCF were pretreated with AG, an inhibitor of AGE production (the aldehyde group of carbohydrates, but not proteins, reacts with the amino group of AG) [8,12,13,14,15], and then exposed to GA. Since AG is a reagent that induces cytotoxicity by inhibiting the activity of enzymes containing a carbonyl group [15], the cell viability decreased only after treatment with 16 mM AG (Figure 1c). However, the viability and intracellular TAGE levels of HCF treated with GA and AG did not significantly differ from those treated with AG only (Figure 1c,d). Therefore, AG completely inhibited the effects of GA in HCF. Moreover, we demonstrated that intracellular TAGE induces cell death in HCF. Based on the previous studies, we considered that death of HCF directly inhibited the role of them more than the suppression of HCF activation or differentiation [3,4,5,6].

We previously reported that intracellular TAGE were generated by hepatocytes [16,17], cardiomyocytes [8], skeletal muscle myoblasts [13], pancreatic islet β-cells [12], pancreatic ductal cells [18], and neuroblastoma cells [19], and are responsible for inducing cell death. Therefore, intracellular TAGE is speculated to promote LSRD via cell death and organ dysfunction [2,20]. However, there is only one report that intracellular TAGE generated in the primary cells and induced their death; the primary cells which were used for this investigation were rat cardiomyocytes [8]. We, therefore, predict that the death or dysfunction of cardiomyocytes forms the underlying cause of cardiac tissue damage and CVD. In contrast, the induction of cell death in normal HCF may result in the opposite response because HCF can function against both cardiomyocytes and cardiac tissue [3,4,5,6]. We consider the death of normal HCF by intracellular TAGE to be an interesting phenomenon compared with that of rat primary cardiomyocytes.

We previously demonstrated that GA-treated rat primary cardiomyocytes produced 12.0 μg/mg of intracellular TAGE, which completely inhibited the beating, and decreased cell viability to 39% [8]. Moreover, 28.7 μg/mg of intracellular TAGE decreased cell viability by 13%. Simultaneous production of intracellular TAGE by both cardiomyocytes and cardiac fibroblasts under high glucose or fructose conditions hinders the ability of cardiac fibroblasts in protecting cardiomyocytes and cardiac tissue, resulting in cell dysfunction, as well as impaired signaling, and death. In addition, the mechanisms involved in the repair of dysfunctional or dying cardiomyocytes and cardiac tissue may be impaired. In contrast, intracellular TAGE may induce cell death in cardiac fibroblasts, leading to the suppression of the undesirable effects on cardiomyocytes and cardiac tissue [3,4,5,6]. 

A novel technique for directly reprogramming cardiac fibroblasts into cardiomyocytes was recently reported [21]. In this, patients with CVD having dysfunctional or dead cardiomyocytes are supplied with new cardiomyocytes. However, intracellular TAGE induces death of the HCF cells, and may potentially interfere with the technique. 

We also demonstrated the effects of extracellular TAGE on normal HCF (Figure 2). Although TAGE is expressed by HCF, a 10- to 30-fold higher concentration of TAGE-BSA, compared to normal physiological levels (7–16 μg/mL), did not induce cell death [12,22]. Collectively, these findings and the results of this study suggest that extracellular TAGE at physiological levels does not induce cell death of the cardiac fibroblasts in patients with CVD. TAGE-BSA has previously been shown to suppress the effects of TGF-β in the human hepatic stellate cell line, LX-2 [23]. Inhibition of TGF-β by extracellular TAGE may suppress the activation and differentiation of cardiac fibroblasts, leading to the desirable/undesirable role of cardiac fibroblasts. Furthermore, the main factors that can activate and differentiate cardiac fibroblasts without TGF-β (e.g. IL-1β, IL-6) were reported [3,4,5,6]. In the future, we will perform the assay by stimulating HFC cells with these factors to assess the effect of extracellular TAGE under physiological conditions. 

The present study has two limitations. First, we did not investigate whether cardiac fibroblasts differentiate into cardiac myofibroblasts in the absence of GA and AG in-vitro. PromoCell GmbH (Land Baden-Wurttemberg, Germany) ensure that the smooth muscle α-actin in HCF cells which we obtained (Cat.C-12375) were negative. However, we were unable to provide data to show that cardiac fibroblasts did not differentiate into cardiac myofibroblasts in this experiment. Second, we did not perform a wound healing test and TGF-β stimulation of HCF cells which generate intracellular TAGE. Since the death of HCF was dramatically induced by 1 mM GA treatment, we considered that the condition was unsuitable for the aforementioned tests. Recently, we have been challenged to develop a high sensitivity slot blot analysis method to detect even minor amounts of intracellular TAGE. Cell viability may be slightly decreased by the minor intracellular TAGE, therefore validating the existence of a significant correlation between minor amounts of intracellular TAGE and cell viability. The wound healing assay, and TGF-β stimulation of HCF cells could be effectively performed under suitable conditions.

Since cell death in cardiac fibroblasts directly and markedly suppresses their role, we only examined cell viability in the present study. 

In conclusion, only intracellular TAGE strongly induced cell death in HCF under physiological conditions, and, thus, may directly suppress the role of HCF, which exert both protective and damaging effects on the heart.

## 4. Materials and Methods 

### 4.1. Reagents and Antibodies

HCF, fibroblast growth medium 3 (customized as a ready-to-use kit with a final glucose concentration of 8 mM), and the Detachi kit were obtained from PromoCell GmbH (Land Baden-Wurttemberg, Germany) (HCF: thawing and seeding in passage 2 (3rd culture), and the number of cell divisions was at least 15, confirmed by PromoCell GmbH). All other reagents and kits were purchased and prepared as previously described [8,12,13,17]. TAGE-BSA and the rabbit polyclonal anti-TAGE antibody were prepared as previously described [24]. 

### 4.2. Viability of Cells Treated with GA only

Cells were seeded on seven wells of a 96-well microplate (1.0 × 10^4^ cells/cm^2^). The incubation of HCF and treatments with 0, 1, 1.5, and 2 mM GA were performed as previously described [13]. To assess cell viability, NADPH activity levels were measured with the WST-8 assay kit, according to the manufacturer’s instructions (Dojindo Laboratories, Kumamoto, Japan).

### 4.3. Viability of Cells Treated with GA and AG

Cells were seeded on seven wells of a 96-well microplate (1.0 × 10^4^ cells/cm^2^). Cells were incubated and treated with 0 and 16 mM AG for 2 h, followed by 0, 1, and 2 mM GA. The assessment of cell viability was performed as described in Section 4.4.

### 4.4. Measurement of Intracellular TAGE Levels in HCF Treated with GA and AG Using a Slot Blot (SB) Analysis

Cells (1.0 × 10^4^ cells/cm^2^) were incubated in a 35-mm dish. Cells (1.0 × 10^4^ cells/cm^2^), incubated in a 35-mm dish were treated with GA and AG at the same concentrations as in the cell viability experiment. After the medium was removed and the cells were washed with PBS (-), the cells were lysed in buffer (2M thiourea, 7 M urea, 4% 3-[(3-cholamido-propyl)-dimethyl-ammonio]-1-propane sulfonate), and 30 mM Tris solution (protease inhibitor solution;9:1)] [8,18]. The SB analysis was performed as previously described [8,18].

### 4.5. Treatment of HCF with NG-BSA and TAGE-BSA and Evaluation of Cell Viability

Cells were seeded on seven wells in a 96-well microplate (1.0 × 10^4^ cells/cm^2^). The conditions of the TAGE-BSA and NG-BSA treatments were as previously described with certain modifications [12,13]. HCF were treated with 0, 100, and 200 μg/mL of NG-BSA and TAGE-BSA, and maintained for 24 h. To assess cell viability, NADPH activity levels were measured using the WST-8 assay kit.

### 4.6. Statistical Analysis 

The cell viability assay and SB analysis were performed in three independent experiments. Stat Flex Software Version 6 was used to perform multiple comparisons (Artech Co., Ltd., Osaka, Japan). Data are shown as the mean ± standard deviation. Significant differences in the means of each group were examined via Tukey’s test after one-way analysis of variance. The statistical significance of differences was set at *p* < 0.05.

## Figures and Tables

**Figure 1 metabolites-12-00615-f001:**
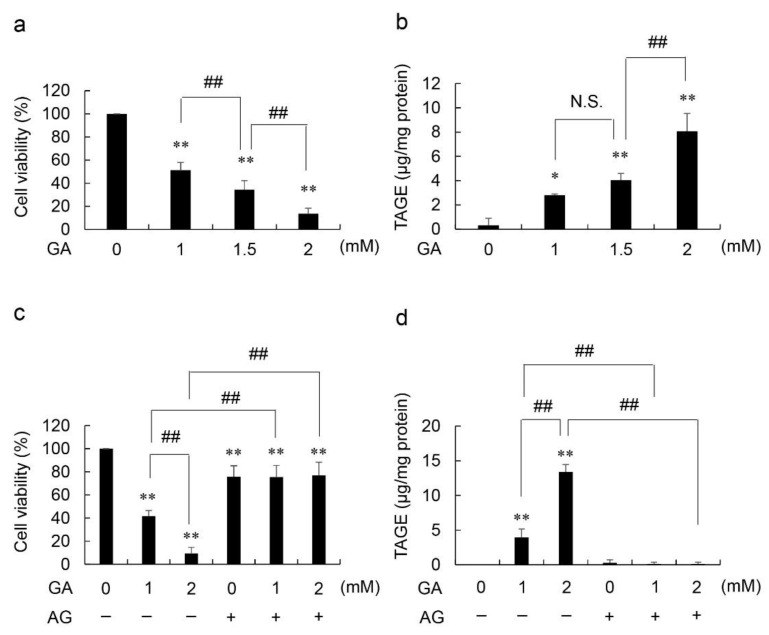
Viability based on NADPH activity and accumulated levels of intracellular TAGE in GA- and AG-treated HCF (1.0 × 10^4^ cells/cm^2^) incubated in fibroblast growth medium-3. (**a**,**b**) effects of treatments with 0, 1, 1.5, and 2 mM GA for 24 h; (**c**,**d**) effects of pretreatments with 0 or 16 mM AG for 2 h, followed by 0, 1, and 2 mM GA for 24 h; (**a**,**c**) NADPH activity was measured to calculate cell viability in four independent experiments. Data are shown as the mean ± S.D (n = 4). (**b**,**d**) The SB analysis was performed to assess the accumulated levels of intracellular TAGE. Data are shown as the mean ± S.D (n = 3). *p*-values were determined using Tukey’s test. (**a**,**b**) * *p* < 0.05 vs 0 mM GA ** *p* < 0.01 vs 0 mM GA. ^##^
*p* < 0.01. N.S.: not significant (**c**,**d**) ** *p* < 0.01 vs 0 mM GA without AG. ^##^
*p* < 0.01.

**Figure 2 metabolites-12-00615-f002:**
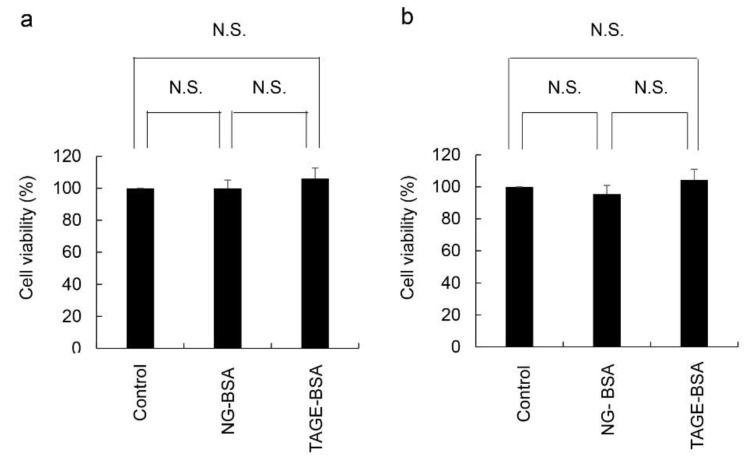
Effects of TAGE-BSA on HCF were evaluated using the NADPH activity assay with the WST-8 method. Control: PBS (−). HCF (1.0 × 10^4^ cells/cm^2^) were incubated in fibroblast growth medium-3. Data are shown as the mean ± S.D (n = 3). P-values were based on Tukey’s test. N.S.: not significant. (**a**) HCF was treated with 100 μg/mL NG-BSA and TAGE-BSA for 24 h. (**b**) HCF was treated with 200 μg/mL NG-BSA and TAGE-BSA for 24 h.

## Data Availability

All data presented in this article are available upon request from the corresponding author.

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
