# Peer review of "Intracellular Toxic Advanced Glycation End-Products May Induce Cell Death and Suppress Cardiac Fibroblasts"

_metabolites, 2022, doi:10.3390/metabo12070615_

Round 1

Reviewer 1 Report

In the revised manuscript, the introduction is slightly clearer. However, the presented data do not seem to support the claims. There are some implications that TAGE blocks TGFb but the authors did not do a TGFb stimulation or any assay to prove it. In the revised manuscript, the authors make new claims about how TAGEs "may suppress fibrotic functions" but no mechanisms or data supports these statements. The only data is on cell viability, not fibrosis. Also no mention on the importance of cardiac fibroblasts.

Author Response

We would like to thank the reviewers for their comments on our previous manuscript (Code: 1769013). 

Point-by-point responses to the comments of Reviewer 1 

In the revised manuscript, the introduction is slightly clearer.However, the presented data do not seem to support the claims.

Comment 1: There are some implications that TAGE blocks TGFb but the authors did not do a TGFb stimulation or any assay to prove it.

Response 1: We rewrote about the relationship between intra-/extracellular TAGE and TGF-β. (i) The relationship between intracellular TAGE and TGF-β is unclear in human cardiac fibroblasts (HCF) cells.

We only provided the data that intracellular TAGE induced the death of HCF cells.

Therefore, we deleted “such as TGF-β” in Line 151 in the Discussion section of the previous manuscript.

The TGF-β stimulation test against HCF cells which generate intracellular TAGE could not be performed in this study due to decreased viability and death of HCF induced by even 1 mM GA treatment, which was considered unsuitable for this test.

Recently, we have been challenged to develop the high sensitivity slot blot analysis method to detect minor intracellular TAGE. The cell viability may be slightly decreased by the minor intracellular TAGE. Thus, if we could prove the existence of a significant correlation between the quantity of the minor intracellular TAGE and the cell viability, the TGF-β stimulation test, and the other assays may be performed under the suitable condition.

We have included this explanation as a study limitation in the Discussion section.

(ii) Although we could not provide the data about the relationship between extracellular TAGE and TGF-β in HCF cells, we reported that TAGE-modified bovine serum albumin (TAGE-BSA, the model of extracellular TAGE) suppressed the effects of TGF-β in the human hepatic cell line LX2 (Ref. 23). Therefore, we described this information in the Discussion section.

Moreover, we predicted that only extracellular TAGE may inhibit the effects of TGF-β in cardiac fibroblasts in the Discussion section of the manuscript.

Comment 2: In the revised manuscript, the authors make new claims about how TAGEs "may suppress fibrotic functions" but no mechanisms or data supports these statements. The only data is on cell viability, not fibrosis.

Response 2: We agree with the comment by Reviewer 1, and rewrote some sentences in the Title, Abstract, and Discussion sections.

In our previous manuscript, we described “Fibrotic function”, “fibrosis function”, “excess or abnormal fibrosis of cardiac tissue”, and “functions which are occurred via fibrosis” (Line 3, 29, 156, 169, and 193). We rewrote these sentences to suggest that intracellular TAGE may suppress the role of cardiac fibroblasts.

Comment 3: Also no mention on the importance of cardiac fibroblasts.

Response 3: In the previous version of our manuscript, we described that cardiac fibroblasts can perform fibrotic healing and cause excess fibrosis in the Introduction section. However, the information on the role of cardiac fibroblasts and the crosstalk between cardiomyocytes and cardiac fibroblasts were not sufficiently explained.    

Therefore, in the current version, we have described that (i) cardiac fibroblasts secrete TGF-β to induce the apoptosis of cardiomyocytes (ii) cardiac fibroblasts secrete connective tissue growth factor (CTGF) to induce antioxidant effects in cardiomyocytes, (iii) cardiac fibroblasts secrete angiotensin II to increase reactive oxygen spices (ROS) via nicotinamide adenine dinucleotide phosphate (NADPH) oxidase in cardiomyocytes.

Cardiac fibroblasts are able to exert opposite role against both cardiomyocytes and cardiac tissue. We described this information in the Abstract, Introduction, and Discussion sections; the number of Ref.3 and 4 were changed accordingly.

Reviewer 2 Report

None

Author Response

We would like to thank the reviewers for their comments on our previous manuscript (Code: 1769103).

Point-by-point responses to the comments of Reviewer 2

Reviewer 2 recommended that our manuscript should be revised for language. Therefore, the revised manuscript and “Point by Point Response to Reviewers” were edited using the English editing service by Editage Inc (Tokyo, Japan).

Moreover, we described in the Acknowledgments section that our manuscript has been revised by Medical English Service Co., Ltd. (Kyoto, Japan) and Editage Inc.

Reviewer 3 Report

Thank you for improving your manuscript. It has become more clear. Main finding, and now, only (!) finding, is the in vitro work showing differences in effects of intracellular vs. extracellular TAGE.

Major comments: 

novel finding is very limited. Additional work that would be interesting is now part of the limitations. Optionally, I would prefer seeing further work on fibroblast differentiation, wound healing assay and others.

Minor comments: 

1) line 86/87 in the manner same... ?? - unclear sentence

2) line 92: you mean extracellular TAGE?

3) line 162ff: too detailed for a discussion. I recommend deletion of:

"However, we provided the data on cardiac fibroblasts (3rd culture), which was obtained from PromoCell GmbH (Cat. 164 C-12375). HCF underwent thawing and seeding in passage 2 (3rd culture). The results of an analysis of smooth muscle α-actin with flow cytometry were negative. Therefore, HCF did not differentiate into cardiac myofibroblasts in the 3 rd culture. Although we need to use fibroblast growth medium 3 (which is customized as a ready-to-use kit) and the Detachi kit to incubate and passage HCF, the instructions did not state whether to use plastic dishes or microplates"

Author Response

We would like to thank the reviewers for their comments on our previous manuscript (Code: 1769103).

 Point-by-point responses to the comments of Reviewer 3

 Comments and Suggestions for Authors

Thank you for improving your manuscript. It has become more clear. Main finding, and now, only (!) finding, is the in vitro work showing differences in effects of intracellular vs. extracellular TAGE.

Major comments:

Comment 1: novel finding is very limited. Additional work that would be interesting is now part of the limitations. Optionally, I would prefer seeing further work on fibroblast differentiation, wound healing assay and others.

Response 1: We agree with the comment by Reviewer 3 regarding the limited novelty of our findings. However, the possibility that intracellular TAGE induces the death of HCF is rarely reported compared with that of other cells. In our previous investigation of intracellular TAGE, there is only one report that intracellular TAGE were generated in the primary cells and induced their death; the primary cells which were used for this investigation were rat cardiomyocytes (Ref. 8). We can predict that the death or dysfunction of cardiomyocytes causes the undesirable phenomenon for cardiac tissue.

In contrast, the induction of cell death in HCF by intracellular TAGE may result in the opposite response because HCF exert opposite roles in both cardiomyocytes and cardiac tissue. We consider that the death of HCF by intracellular TAGE is an interesting phenomenon compared with that of rat primary cardiomyocytes.

We described this information and suggestion in the Discussion sections.

On the other hand, we understand that we should further research fibroblast differentiation, using wound-healing assays and other assays in the future.

However, wound healing assay against HCF cells which generate intracellular TAGE was not feasible in this study due to the death of HCF by 1 mM GA treatment.

Recently, we have been challenged to develop the high sensitivity slot blot analysis method to detect minor intracellular TAGE. The cell viability may be slightly decreased by minor intracellular TAGE. If we can prove that a significant correlation exists between the quantity of the minor intracellular TAGE and the cell viability, the wound healing assay and other examinations (e.g. TGF-β stimulation test against HCF cells) may be performed under suitable conditions.

We included this information as a study limitation in the Discussion section.

Though we could not provide the data about the relationship between extracellular TAGE and TGF-β in the cardiac fibroblasts, we reported that TAGE-modified bovine serum albumin (TAGE-BSA, the model of extracellular TAGE) suppressed the effects of TGF-β in the human hepatic cell line LX2 (Ref. 23). Therefore, we described this information in the Discussion section.

Moreover, we predicted that extracellular TAGE may inhibit the effects of TGF-β in cardiac fibroblasts and included it in the Discussion section.

Minor comments:

Comment1: line 86/87 in the manner same... ?? - unclear sentence

Response 1: We rewrote these sentences.

Comment 2: line 92: you mean extracellular TAGE?

Response 2: We wanted to report on the examination of extracellular TAGE. Therefore, we rewrote the subheading and sentences to enhance clarity

Comment 3: line 162ff: too detailed for a discussion. I recommend deletion of:

"However, we provided the data on cardiac fibroblasts (3rd culture), which was obtained from PromoCell GmbH (Cat. 164 C-12375). HCF underwent thawing and seeding in passage 2 (3rd culture). The results of an analysis of smooth muscle α-actin with flow cytometry were negative. Therefore, HCF did not differentiate into cardiac myofibroblasts in the 3 rd culture. Although we need to use fibroblast growth medium 3 (which is customized as a ready-to-use kit) and the Detachi kit to incubate and passage HCF, the instructions did not state whether to use plastic dishes or microplates"

Response 3: We agree with the comment by Reviewer 3. However, we included these sentences as suggested by Reviewer 1. We have rephrased and shortened the sentences in this version of the manuscript.

Round 2

Reviewer 1 Report

Comments addressed

This manuscript is a resubmission of an earlier submission. The following is a list of the peer review reports and author responses from that submission.

Round 1

Reviewer 1 Report

The authors are studying the effect of intracellular toxic advanced glycation end-products (TAGE) on human cardiac fibroblasts, which have been previously shown to cause the dysfunction and cell death of cardiomyocytes. The manuscript and lack of data presented makes the paper weak and unconvincing argument that cardiac fibroblasts are responsible for secretion of TAGEs, and does not provide any data on the interaction between cardiac fibroblasts and cardiomyocytes or new insight on how TAGE are secreted. Specific comments can be found below:

Abstract: The abstract is unclear as to the cause and effect of these TAGE, are they secreted by the cardiac fibroblasts? Are they circulating at higher levels just due to the disease? If there is secretion is this not more intercellular or paracrine signaling instead of intracellular? The connection between fibrotic healing and cell death of cardiomyocytes is also unclear, we would suggest making a more clear connection between the response of cardiomyocytes to cardiac fibroblast signaling. 

Line 16: Saying the TAGE promote LSRD seems indirect, how do they induce cardiovascular disease?

Line 21: Spelling and grammar error - “cardiaomyocytes” should be “cardiomyocyte”

Line 22: Should read “With anti-TAGE antibody staining”

Introduction: The transition of fibroblasts to myofibroblasts does not always protect cardiac tissue as cited in line 39, and can often have deleterious effects. 

Line 37: Should read “differentiation” instead of differentiate

Line 42 and 43: Both instances should read “reactions” not reaction

Results: How were sections determined to be “non-cardiomyocyte” areas? You discuss in the discussion why pretreatment with aminoguanidine prevents effects of GA but it is not clear that it inhibits AGE production in the results.

Line 60: Should read “non-cardiomyocyte” instead of non-cardiomyocytes

Line 61: Should read “were stained with”

Line 76: Should read “of” instead of f

Line 84: Should read “the control” instead of control, “saline” instead of salline

Discussion: It is rather unconvincing that the TAGE are generated and secreted by the cardiac fibroblasts based on this data. The activity of TGF-β is not always advantageous to the heart, and can often further disease. Why were studies not done to probe if extracellular TAGE inhibited activation of fibroblasts to myofibroblasts by TGF-β?

Line 105: Should read “bred under special conditions such as a high glucose or fructose diet” instead of bread in the especially condition 

Line 141: This sentence does not make much sense

Line 161: Important theme of what?

Line 162: This sentence directly contradicts the argument made in the abstract

Line 182: Should read “donor” not donner

Methods: Were fibroblasts seeded directly onto plastic microplates? Because this stiffness induces transition into myofibroblasts already, which can affect the results of the study.

Line 207: “of them” should be removed

Reviewer 2 Report

Takata et al investigated the intracellular advanced glycation 15 end-products (AGEs) as toxic AGEs (TAGE) was generated in the non-cardiaomyocytes areas in the cardiac tissue of the 21 Wistar/ST rats and they also found that intracellular TAGE induced the cell death of human cardiac fibroblasts (HCF) under 26 physiological conditions.

The experimental approaches used in this brief report are not very strong except for the cell viability assay.

In Figure1, the authors showed non-cardiomyocyte areas in the cardiac tissues of Wistar/ST rats. Why Wistar/ST rats? Did the authors test GA or AG via an IV or IP on the mice prior to this experiment? If not, suggest the following experiment:

  1. Treat mice with GA or/and AG and check TAGE in both cardiac and non-cardiac tissue.
  2. Isolated cardiomyocytes and fibroblasts from Wistar/ST rats and repeat the series of HCF experiments

Figure2, 2 mM of GA treatment for 24 hrs showed a very significant effect. It would be better to screen the effective concentration at different time points.

Reviewer 3 Report

Takata et al. claim that intracellular AGEs induce cell death and suppresses fibrotic healing.

To my understanding, none of which is shown with the presented data: first, the authors claim to be able to identify fibroblasts by H&E staining which is doubtful. H&E stains not only cardiomyocytes but also fibroblasts. Additionally, immunohistochemical AGE staining seems mostly positive in cardiomyocytes (Figure 1b). Second, wound healing was not tested in this manuscript.

To show fibroblast AGE, co-staining with FSP-1 or another fibroblast marker is necessary; or non-overlay with a cardiomyocyte marker. 

To test wound healing, an injury model in vivo or at least a simplified "wound healing" in vitro assay should be used.

Many minor comments:

The second sentence of the results part seems to be the vital result but the sentence is gramatically incorrect and not understandable. 

Figure Legend (Fig.1) is unclear: n=5 per group? Images shown are mostly represent cardiomyocytes, not "non-cardiomyocytes". 40x should be shown for a major claim that cannot be confirmed with the data presented.

Known (and unknown) effects of AG are not explained. Known and (unknown) effects of NG-BSA are not explained. Why was AG and NG-BSA used? To prove what? In general, a more focused introduction on AGEs seems necessary: are there non-toxic AGEs? What is their physiological function? 

A graphical abstract should be included.

line 76: "... f ..." ?

English needs major revision to understand presented data.